Influence of SPIO labelling on the function of BMSCs in chemokine receptors expression and chemotaxis

Liu Yuanchun 15372048@qq.com
Huang Wanyi
Wang Huiyang
Lu Wei
Guo Jiayu
Yu Li
Wang Lina eywangln@scut.edu.cn
Department of Pediatrics, Guangzhou First People’s Hospital, School of Medicine, South China University of Technology , Guangzhou , China
Sonowal Himangshu
Electronic publication date: 2023 Jun 2
Publication date: 2023
Volume: 11
Electronic Location ID: e15388
Received 2022 Oct 10; Accepted 2023 Apr 19
Copyright: ©2023 Liu et al.
Copyright year: 2023
Copyright holder: Liu et al.
License: This is an open access article distributed under the terms of the Creative Commons Attribution License, which permits unrestricted use, distribution, reproduction and adaptation in any medium and for any purpose provided that it is properly attributed. For attribution, the original author(s), title, publication source (PeerJ) and either DOI or URL of the article must be cited.
License URL: https://creativecommons.org/licenses/by/4.0/

Keywords: SPIO, Chemokine receptor, Chemotaxis, aGVHD, BMSC

Funding: Science and Technology Projects in Guangzhou 202102080130 This work was funded by the Science and Technology Projects in Guangzhou (item number: 202102080130). The funders had no role in study design, data collection and analysis, decision to publish, or preparation of the manuscript.

==============================
Bone marrow-derived mesenchymal stem cells (BMSCs) are increasingly being used in bone marrow transplantation (BMT) to enable homing of the allogeneic hematopoietic stem cells and suppress acute graft versus host disease (aGVHD). The aim of this study was to optimize the labelling of BMSCs with superparamagnetic iron oxide particles (SPIOs), and evaluate the impact of the SPIOs on the biological characteristics, gene expression profile and chemotaxis function of the BMSCs. The viability and proliferation rates of the SPIO-labeled BMSCs were analyzed by trypan blue staining and CCK-8 assay respectively, and the chemotaxis function was evaluated by the transwell assay. The expression levels of chemokine receptors were measured by RT-PCR and flow cytometry. The SPIOs had no effect on the viability of the BMSCs regardless of the labelling concentration and culture duration. The labelling rate of the cells was higher when cultured for 48 h with the SPIOs. Furthermore, cells labeled with 25 µg/ml SPIOs for 48 h had the highest proliferation rates, along with increased expression of chemokine receptor genes and proteins. However, there was no significant difference between the chemotaxis function of the labeled and unlabeled BMSCs. To summarize, labelling BMSCs with 25 µg/ml SPIOs for 48h did not affect their biological characteristics and chemotaxis function, which can be of significance for in vivo applications.

Introduction

Allogeneic hematopoietic stem cell transplantation (allo-HSCT) is known to improve the outcomes of patients with hematological malignancies (Copelan et al., 2013; Merli et al., 2019). For instance, haploidentical allotransplantation is an effective and safe treatment option for acute myeloid leukemia, aplastic anemia and myelodysplastic syndrome. HCT is also a viable option for the relapse cases. Allo-HSCT has been successfully used to treat Philadelphia chromosome-negative acute lymphoblastic leukemia (Chang et al., 2017; Xu et al., 2017; Zeng et al., 2021; Fan et al., 2019; Fang et al., 2022). However, it is beset with several challenges such as the lack of suitable donors (Atilla, Ataca Atilla & Demirer, 2017; Zhu et al., 2018) and induction of acute graft-versus-host disease (aGVHD) (Aladağ, Kelkitli & Göker, 2020; Naserian et al., 2020). Transplantation of ex vivo-expanded bone marrow-derived mesenchymal stem cells (BMSCs) can obviate the disadvantages of allo-HSCT. A systematic review and meta-analysis showed that BMSCs infusion can reduce the incidence of chronic graft-versus-host disease (cGVHD) and relieve the symptoms of aGVHD. Furthermore, the infusion of umbilical mesenchymal stem cells (U-MSCs) and BMSCs after HSCT reduced the incidence of cGVHD and promoted engraftment, which decreased relapse and death rates (Zhao et al., 2019). In addition, a case report described that transplantation of human bone marrow-derived mesenchymal stem cells (hBMSCs) treated steroid-refractory gastrointestinal aGVHD successfully without complications in a 15-year-old boy with acute lymphoblastic leukemia (Moritani et al., 2019). BMSCs were first identified by Friedenstein, Chailakhyan & Gerasimov (1987) as fibroblast colonies formed by bone marrow explants. Former studies show that BMSCs can promote the homing of transplanted bone marrow cells through specific chemokines and their receptors, such as CXCL1 (Wang et al., 2021), CXCL12, CCL21 and CCL27, and augment hematopoiesis by secreting multiple cytokines such as IL 6, IL 10, TGF- β (Jin et al, 2018; Heirani-Tabasi et al., 2018; Hocking, 2015; Mohammadali, Abroun & Atashi, 2018; Kim et al., 2015; Köse et al., 2018; da Silva et al., 2022; Guo et al., 2018a; Hastreiter et al., 2021; Sun et al., 2021). Moreover, the BMSCs have been shown to mitigate severe, steroid-refractory aGVHD in the liver, skin and lungs, although the exact mechanisms are unclear (Song et al., 2015a; Lee et al., 2015; Auletta et al., 2015; Bonig et al., 2019). Tracing BMSCs in real time may help elucidate the mechanisms underlying the above observations. BMSCs can be labeled with BrdU, fluorescein, green fluorescent protein, and chromosomal and magnetic markers (Munoz et al., 2005; Dumitriu et al., 2001; Segers et al., 2006; Raimondo et al., 2006). Magnetic particles are retained for a long duration after intravenous injection, and are ideal for tracking the homing, implantation and functions of BMSCs in real time in situ (Urbán et al., 2008), especially as therapeutic agents. For instance, BMSCs labeled with superparamagnetic iron oxide particles (SPIOs) can be monitored in vivo by non-invasive magnetic resonance imaging (MRI). In an animal model, scientists labeled blastema cells with SPIOs and tracked the cells successfully for a duration of 48 days in MRI (Lauridsen et al., 2018). SPIOs have the advantages of non-toxicity, prolonged retention and stable excretion (Guo et al., 2018b; Will et al., 2015; Rosenberg et al., 2016). Most studies on labeled BMSCs have been focused on determining the toxicity of the labelling particles and optimizing the concentration for real-time in vivo imaging (Liu et al., 2016; Rosenberg et al., 2013; Roeder et al., 2014). Several studies have suggested that the SPIOs with different physico-chemical characteristics are toxic to cells at concentrations greater than 100 µg/ml (Singh et al., 2010; Yildirimer et al., 2011; Palacios-Hernandez et al., 2020; Marcus et al., 2016). Likewise, chondrocytes labeled with small iron oxide particles show aberrant expression of differentiation-related genes (Foldager et al., 2011). However, little is known regarding the influence of SPIOs and other magnetic particles on the chemokine gene expression and chemotaxis function of BMSCs, which is critical to their clinical role in allo-HSCT and aGVHD.

The aim of our study was to determine the impact of different concentrations and labelling durations of SPIOs on the biological characteristics of the BMSCs. We chose rat BMSCs (rBMSCs) from SD rats for our study, which the physiological characteristics meet the requirements of experimental purposes and the needs of subsequent animal experiments. We found that SPIOs can effectively label rBMSCs without affecting their viability and chemotactic functions. The SPIOs-labeled BMSCs can be used to elucidate the mechanisms underlying their functions in BMT and aGVHD through real-time in vivo monitoring.

Materials and Methods

Isolation and culture of rat BMSCs (rBMSCs)

SD male rats aged 4–5 weeks old and weighing about 160–200 g obtained from the Animal Experiment Center of Southern Medical University were euthanized by intravenous injection and dissected to expose the femur, and the study was approved by the Institutional Animal Care and Use Committee of Guangzhou First People’s Hospital. The euthanizing criteria was according to our former “Vertebrate Animal Study Approval” by the ethics committee on experimental animals. The ends of the femurs were cut with a pair of bone forceps, and the bone marrow was flushed out using a 27-gauge needle attached to a 10 ml syringe containing alpha-DMEM/F12 (1:1) medium. The single cell suspension was seeded in a 25 cm2 flask and cultured in alpha-DMEM/F12 supplemented with 10% fetal bovine serum (FBS; Ausgenex, FBS500) and 1% penicillin/streptomycin at 37 ° C under 5% CO2 in a humid incubator (Thermo Fisher Scientific, Waltham, MA, USA). The culture medium and the non-adherent cells were gently removed 24 h later, and fresh medium was added. Thereafter, the cell medium was changed every 2–3 days and the cells were passaged at 90% confluency. The primary isolated rBMSCs were defined as passage 0. Cells from passages <10 were used for the experiments. The methodology is outlined in Fig. 1.

Figure 1 Schematics of the experimental design.

The rBMSCs were cultured with complete alpha-DMEM/F12 with 0, 25, 50 or 75 µg/ml SPIOs for 24, 48 or 72 h. The expanded rBMSCs were identified by flowcytometry and the labelling rate, proliferation and viability were measured using suitable assays. The expression of different chemokine receptors was analyzed by RT-PCR and flow cytometry. Transwell assay was used to evaluate the migration rates of the BMSCs.

The rBMSCs were cultured with complete alpha-DMEM/F12 with 0, 25, 50 or 75 µg/ml SPIOs for 24, 48 or 72 h. The expanded rBMSCs were identified by flowcytometry and the labelling rate, proliferation and viability were measured using suitable assays. The expression of different chemokine receptors was analyzed by RT-PCR and flow cytometry. Transwell assay was used to evaluate the migration rates of the rBMSCs.

Flow cytometry

The primary rBMSCs were washed twice with PBS, re-suspended at the density of 1 × 106/ml, and incubated with: anti-CD29 (Biolegend Cat # 102221, RRID: AB_528789), anti-CD34 (Santa Cruz Cat # sc-7324, RRID: AB_2291280), anti-CD44 (BD Cat # 553456, RRID: AB_10515282) and anti-CD45 (Biolegend Cat # 202216, RRID: AB_1236411) antibodies at 37 °C in the dark for 30 min. The stained cells were washed with PBS and re-suspendered in 200 µl PBS (n = 3). The rBMSCs co-cultured with SPIOs for 24 h, 48 h and 72 h were trypsinized and centrifuged at 1,500 rpm for 15 min, and re-suspended in PBS at the density of 1 × 106 cells/tube. The aliquots were incubated with anti-CXCR-4 (Santa Cruz cat. no. sc-53534, RRID: AB_782002), anti-CXCR-7 (R&D Cat # FAB8399RG, RRID: AB_2917964), anti-CCR-10 (R&D Cat # FAB2815A-025, RRID: AB_1151964) and anti-CXCR-3 (R&D Cat # FAB8109P, RRID: AB_2917963) antibodies for 30 min at room temperature in the dark. The stained cells were washed and re-suspended in 200 µl PBS. Unstained controls were also included. All samples were analyzed by setting appropriate side and forward scatter gates to identify the rBMSCs cell population. The cells were acquired in the BD FACSCANTO Plus 10C flow cytometer (BD Biosciences, Franklin Lakes, NJ, USA) (n = 7), and the expression of the markers was analyzed by Flowjo software (version Flow Jo X 10.0.7r2; FlowJo LLC, Ashland, OR, USA).

Cell labelling assay

The rBMSCs were cultured till 50% confluent, and incubated with different concentrations (25, 50, 75 and 100 µg/ml) SPIOs (Aladdin, cat. no. I140464) for 24 h. The medium was discarded and fresh medium lacking SPIOs was added. After culturing for another 24 h, 48 h and 72 h, the labeled cells were harvested for subsequent tests (n = 3).

Prussian blue staining

The labeled rBMSCs were washed thrice with PBS and fixed with 4% paraformaldehyde for 30 min. After discarding the paraformaldehyde, the cells were incubated with 2% potassium ferrocyanide in 6% hydrogen acid at room temperature for 30 min. The stained cells were washed thrice with Hank’s buffer (Corning, cat. no. 21-022-CVR), counterstained with nuclear fast red for 2–5 min, and washed again. The number of positively-stained cells were counted under a microscope at 100 × magnification, and the percentage of SPIOs-labeled cells to the total number of cells were calculated (n = 3).

Trypan blue staining

Labeled rBMSCs were suspended in PBS at the density of 1 × 106 cells/ml, and 90 µl cell suspension was mixed with 10 µl 0.4% trypan blue solution. The cells were counted within 3 min of staining, and the viability was calculated as: number of unstained cells/total cells × 100% (n = 3).

CCK-8 assay

The rBMSCs were seeded in 96-well plates at the density of 2 × 104/well and cultured overnight. The medium was discarded the following day, and fresh medium containing different concentrations of SPIOs was added. After culturing for 24 h, 48 h and 72 h, the medium was replaced with 100 µl fresh medium. CCK-8 reagent was added to each well and the cells were incubated for 2 h in the dark. The absorbance (OD) value at 490 nm was measured using a microplate reader (BioTek, Winooski, VT, USA) (n = 5).

Quantitative reverse transcription-polymerase chain reaction

The total RNA was extracted using TRIZOL (cat. no. 15596-026; Thermo Fisher Scientific, Waltham, MA, USA) according to the manufacturer’s instructions, and reverse transcribed to cDNA using PrimerScript RT Master Mix (Takara, Tokyo, Japan). RT-PCR was performed on the IQ5 System (Bio-Rad) using SYBR Green Mastermix (Takara, Tokyo, Japan) (n = 3). Each sample was analyzed in duplicate and the relative expression of the target genes was normalized to β-actin. The primer sequences are listed in Table 1. The conditions for telomere amplification and HGB gene amplification were as follows: 95 °C for 30 min, followed by 40 cycles of 95 °C for 15 s, and 60 °C for 1 min.

Transwell assay

Complete DMEM/F12/F12 (10% FBS) was dispensed into transwell inserts with pore size of 8 µm (Corning cat. no. 3422, Costar, Cambridge, MA, USA) in 24-well plates, and incubated overnight. After discarding the medium, control or SPIO-labeled rBMSCs were seeded into the upper chamber of the inserts in complete medium at the density of 8 × 104 cells/200 µl, with or without chemokine receptor antagonists TAK799, Maraviroc or LY294002. The lower chambers were filled with medium supplemented with the corresponding chemokines i.e., CXCL-10, CCL-4 and CCL-19. After incubating the cells for 48 h, the inserts were removed and the cells remaining on the upper surfaces of the membrane were scraped off. The membranes were then washed twice with PBS, fixed in 4% paraformaldehyde for 10 min, and stained with 0.5% crystal violet for 5 min. The migrated cells on the lower surface of the membranes were counted in three random fields per well under a microscope at low magnification (n = 3).

Statistical analysis

Statistical product and service solutions (SPSS) 20.0 software (IBM, Armonk, NY, USA) was used for the statistical analysis. The enumeration data were expressed as mean ± standard deviation. Multiple groups were compared by one-way analysis of variance (ANOVA) followed by Tukey’s post hoc test, or rank sum test followed by Kruskal-Wallis post hoc test. P < 0.05 was considered statistically significant.

Table 1 Primer sequences for rever transcription-quantitative polymerase chain reaction analysis.

Gene	Forward(5′–3′)	Reverse(5′–3′)	
CXCR4	TGCCATGGAAATATACACTTCGG	TGCCCACTATGCCAGTCAAG	
CCR10	GGTGGCTGTGCTGGGTTTGG	GGAGGTGGGAGATCGGGTAGTTC	
CXCR3	GCCAGTCCTCTACAGCCTCCTC	ACAGCCAGGTGGAGCAGGAAG	
CXCR5	GAACTCCCCGATATCGCTAGAC	TGGCCAGTTCCTTGTACAGAT	
CXCR7	CCGCGAGGTCACTTGGTT	CAGTGTGTGTCGTAGCCTGT	
IL-6	GCCCACCAGGAACGAAAGTC	TGGCTGGAAGTCTCTTGCGG	
IL-11	CTTCAGACCCTCGTGCAGAT	CAGGAAGCTGCAAAGATCCCA	
Notes.

CXCR4 CXC-chemokine receptor CXCR4

CCR10 CC chemokine receptor CCR10

CXCR3 CXC-chemokine receptor CXCR3

CXCR5 CXC-chemokine receptor CXCR5

CXCR7 CXC-chemokine receptor CXCR7

IL-6 interleukin-6

IL-11 interleukin-11

Results

Isolation and characterization of rBMSCs

The primary rBMSCs were adherent after overnight incubation, and consistently exhibited a uniform fibroblast-like appearance from passage 0 (Figs. 2A and 2B) to passage 10 (Figs. 2C and 2D) at 50 × and 100 × magnification as indicated. To confirm that the cultured and expanded cells were indeed rBMSCs, we stained the cells for CD markers based on previous reports. As shown in Fig. 3, the isolated rBMSCs were CD29+ (96.8%), CD44+ (99.3%), CD34- (99.2%) and CD45- (98.7%), which is consistent with previous reports and therefore suggested that the isolated and expanded cells were rBMSCs (Dominici et al., 2006; Fiuza-Luces et al., 2016).

Figure 2 The morphology of the rBMSCs.

The rBMSCs displayed a fibroblast-like appearance throughout culture. Images show rBMSCs from passages (A, C) 0 and (B, D) 10 at 50 × and 100 × magnification as indicated.

Figure 3 Identification of rBMSCs.

Flow cytometry plots showing percentages of (A) CD29+, (B) CD44+, (C) CD34- and (D) CD45- cells.

The rBMSCs displayed a fibroblast-like appearance throughout culture. Images show rBMSCs from passages (A, C) 0 and (B, D) 10 at 50 × and 100 × magnification as indicated.

Flow cytometry plots showing percentages of (A) CD29+, (B) CD44+, (C) CD34- and (D) CD45- cells.

rBMScs were optimally labeled with SPIOs after 48 h incubation

The rBMSCs were incubated with different concentrations of SPIOs for varying durations, and the presence of SPIOs was detected by Prussian blue staining. As shown in Fig. 4A, cells incubated with SPIOs had dense blue-stained iron particles in their cytoplasm and the intensity of the color deepened with increasing concentrations of SPIOs. In contrast, no blue-stained iron particle was observed in the control group (Fig. 4A). Furthermore, the percentage of SPIO-labeled cells increased from 54.67% ± 1.15% after 24 h to 98% ± 1% after 48 h of incubation with 50 µg/ml SPIOs. The percentages of the rBMSCs labeled after 48 h of incubation with 25, 50 and 75 µg/ml SPIOs were 90% ± 1.73%, 98% ± 1% and 96% ± 1% respectively, which were consistently higher than that for the other incubation times. (Fig. 4B).

Figure 4 Labelling rate of rBMSCs.

(A) Representative images showing Prussian blue-stained iron particles in the rBMSCs labeled with different concentrations of SPIOs for varying durations. (B) Labelling rate in the indicated groups. (C) Representative images showing live cells in the indicated groups after trypan blue staining. (D) Viability rates in the indicated groups. (E) Proliferation rates in the indicated groups as measured by CCK-8 test. Data are individual means or the mean ± SD of each group from three separate experiments. ∗p < 0.05, ∗∗p < 0.01, ∗∗∗p < 0.001.

SPIOs had no detrimental effect on in-vitro viability and proliferation of rBMSCs

The possible effect of the SPIOs on the viability of rBMSCs were evaluated by trypan blue dye exclusion test, which can help distinguish the dead cells from the live cells. As shown in Figs. 4C and 4D, the extent of trypan blue dye exclusion ranged from 91% ± 3.60% to 98.67% ± 0.58% across the different concentrations of SPIOs, and were not significantly different between the groups. We also used the CellCountKit-8 (CCK-8) assay to evaluate the cell proliferation ability. CCK-8 assay further showed that cells labeled with 25 µg/ml SPIOs had a higher OD value than the cells incubated with higher doses of SPIOs. In addition, the OD value of cells incubated with SPIOs for 72 h was higher than 24 h and 48 h for that the number of cells increases gradually as time went by (Fig. 4E).

SPIOs-labeled rBMSCs expressed higher levels of chemokine receptors and cytokines

We next analyzed the expression levels of several chemokine receptor genes involved in the chemotaxis of SPIOs-labeled rBMSCs. As shown in Fig. 5A, the control and 25 µg/ml SPIOs-labelled cells expressed consistently high levels of CCR5, CCR10, CXCR3, CXCR5, IL 6 and IL11 mRNAs after 48 h of incubation. Furthermore, cells incubated with 25 µg/ml SPIOs for 48 h showed significant upregulation of CCR5 and CXCR5 compared to the 72 h group for the same concentration (p < 0.05), and of CCR10 and CXCR3 compared to cell incubated with 50 µg/ml SPIOs for 24 h. Likewise, the expression of IL-6 and CXCR7 genes were also upregulated in the 25 µg/ml SPIOs/48 h group compared to the control and 25 µg/ml SPIOs/24 h group, and IL-11 was upregulated compared to that in the 50 µg/ml SPIOs/48 h group (Fig. 5B).

Figure 5 SPIO labelling upregulated chemokine receptor and cytokine genes in rBMSCs.

(A, B) The expression level of CCR5, CCR10, CXCR3, CXCR5, IL11, IL 6 and CXCR7 in the indicated groups. β-actin was used as reference internal control to compare the levels of gene expression. Different colors correspond to the labelling concentrations (25/50/75 µg/ml) and culture durations (24/48/72h). Data are individual means or the mean ± SD of each group from three separate experiments. ∗p < 0.05, ∗∗p < 0.01, ∗∗∗p < 0.001.

Furthermore, we also analyzed the levels of the chemokine receptors in the SPIO-labeled and unlabeled BMSCs by flow cytometry. The percentage of CXCR3+ cells in the rBMSCs treated with 25 µg/ml SPIOs for 24, 48 and 75 h were 26.58% ± 3.68%, 71.02% ± 9.93 and 58.31% ± 13.09% respectively, and that of CCR10+ cells at the respective time points were 25.43% ± 9.49%, 43.09% ± 6.68 and 20.72% ± 1.89%. The unlabeled control cells also showed a higher expression of CXCR3 after 48 h, whereas the percentage of CCR10+ cells were unaffected by the duration of culture. There was no significant difference in the percentage of CXCR3+ or CCR10+ cells between the control and 25 µg/ml SPIOs/48 h groups (Figs. 6A–6C). The percentage of CXCR7+ cells in the control group after 24, 48 and 72 h of culture were 67.79% ± 7.77%, 68.59% ± 6.91 and 53.02% ± 8.82% respectively, which were significantly higher than in the 25 µg/ml SPIOs-labeled group at each time point (p < 0.001; Fig. 5E). Nevertheless, rBMSCs labeled with 25 µg/ml SPIOs for 48 h had the highest percentage of CXCR7+ cells (30.03% ± 9.38%) compared to those labeled with higher concentrations of SPIOs for the same duration. No significant difference was observed between the different SPIO-labeled groups incubated for 24 h or 72 h.

Figure 6 SPIO labelling increased percentage of rBMSCs expressing chemokine receptors.

(A) Flow cytometry plots showing percentage of CXCR3/CCR10 control and SPIOs-labeled rBMSCs. (D) Flow cytometry plots showing percentage of CXCR7/CXCR4 control and labeled rBMSCs. Different colors of the bars on top right correspond to the labelling concentrations of SPIOs. (B) CXCR3 expression in different groups. (C) CCR10 expression in different groups. (E) CXCR7 expression in different groups. (F) CXCR4 expression in different groups. Data are individual means or the mean ± SD of each group from three separate experiments. ∗p < 0.05, ∗∗p < 0.01, ∗∗∗p < 0.001.

The expression level of CXCR4 was the highest amongst all chemokine receptors in both the control and SPIOs-labeled groups. The percentage of CXCR4+ cells in the control and 25 µg/ml SPIOs-labeled groups were 94.81% ± 1.72% and 93.47% ± 3.41% respectively after 24 h of culture, and were higher than that in the 50/75 µg/ml SPIOs-labeled groups for the same duration (p < 0.001). In contrast, the cells incubated with 25 µg/ml SPIOs for 48 h or 72 h had a higher proportion of CXCR4+ cells compared to the control (p < 0.01 at 48h) and other labeled groups (p < 0.05 vs 75 µg/ml SPIOs-labeled group at 72h). Thus, CXCR4 expression was higher in cells treated with 25 µg/ml after 48 h and 72 h compared to that in the other groups.

SPIOs did not affect the in-vitro migration and chemotactic function of rBMSCs

To determine the effect of the SPIOs on the chemotactic migration of rBMSCs in-vitro, we cultured the control or 25 µg/ml SPIOs-labeled cells in a transwell system in the presence or absence of the CXCL10, CCL4 and CCL19 antagonists, i.e., TAK 799, MARAVIORIC and LY294002 respectively, and added the respective chemokines to the lower chambers (Fig. 7A). After 48 h, the number of rBMSCs that had migrated to the other side of the transwell membrane was counted in each group. The migration rates of both the control and SPIOs-labeled cells were significantly higher in the absence of the chemokine antagonists (p < 0.001). The percentage of migrated cells per field in response to CXCL10, CCL4 and CCL19 were 81.67 ± 7.36%, 82.00 ±  6% and 69.33 ± 3.05% respectively in the control group, and 74.67 ± 2.51%, 82 ± 2% and /68.33 ± 1.52% in the SPIOs-labeled group. The migration rates of the control group decreased to 29.33 ± 4.04%, 18.00 ± 3% and 20.00 ±  4% in the presence of TAK 799, MARAVIORIC and LY294002 respectively, and that of the SPIOs-labeled cells decreased to 32.33 ± 2.08%, 27.33 ± 2.51% and 29.67 ± 2.08% (Fig. 7B). There was no significant difference in the migration rates of the control and SPIOs-labeled rBMSCs in any of the conditions, indicating that the SPIOs had no effect on the migration and chemotactic behavior of the rBMSCs.

Figure 7 SPIOs labelling has no influence on the migration of rBMSCs.

(A) Representative images of the rBMSCs cultured in transwell chambers in the presence or absence of TAK 799, MARAVIORIC and LY294002 with CXCL10, CCL4 or CCL19 in the lower chamber (magnification 10 ×). (B) Migration rates in the indicated groups. Data are expressed as the mean ± SD of each group (n = 3). The colors of the bars on top right correspond to the number of migrating cells. ∗p < 0.05, ∗∗p < 0.01, ∗∗∗p < 0.001.

Discussion

BMSCs are routinely used to improve implantation and inhibit GVHD during bone marrow transplantation. However, the underlying mechanisms are unclear. Real time monitoring of BMSCs after bone marrow transplantation can help track their homing trajectory to hematopoietic or GVHD-prone organs. For instance, BMSCs labeled with SPIOs can be dynamically monitored by MRI. The aim of this study was to evaluate the effects of SPIOs labeling on the biological and chemotactic functions of rBMSCs. We found that rBMScs were optimally labeled with SPIOs after incubating for 48 h, and the SPIOs did not affect their in-vitro migration and chemotactic function.

We found that labeled rBMSCs with SPIOs did not affect their viability or chemotactic functions. The cells were optimally labeled when incubated with 25 µg/ml SPIOs for 48 h, which increased the expression levels of chemokines and chemokine receptors, and therefore maintained their chemotactic migration in-vitro.

A previous study showed that labeled BMSCs with ferucarbotran, a contrast agent consisting of iron oxide microparticles, decreased their migration ability in a dose-dependent manner (Will et al., 2015). We labeled the rBMSCs with a lower concentration of SPIOs to achieve an efficient labelling rate without compromising on cell viability. This is the first study to show that rBMSCs incubated with 25 µg/ml SPIOs for 48 h can be effectively labeled, while maintaining viability and migration ability in-vitro. This finding is of clinical significance since SPIOs-labeled BMSCs can be tracked in vivo to determine their role in the homing of allo-hematopoietic stem cells and the mitigation of aGVHD. The latter is the main life-threatening complication of allo-HSCT. Several clinical and animal model studies have shown that transplantation of BMSCs improve the outcomes of allo-HSCT (Aladağ, Kelkitli & Göker, 2020; Song et al., 2015b; Fisher et al., 2019). Zhao et al. (2019) conducted a meta-analysis and found that allogeneic BMSCs can be used for the prophylaxis and treatment of aGVHD (Jiang et al. 2022). Therefore, our results can be helpful in understanding the homing trajectory of BMSCs and their ability to facilitate hematopoietic implantation and suppress GVHD, which are the likely basis of their clinical benefits.

The SPIOs-labeled rBMSCs expressed higher levels of different chemokine receptors compared to the unlabeled controls. While CXCR4 mRNA was upregulated in the labeled cells, the percentage of cells expressing CXCR4 protein were similar across all groups. Previous studies have shown that the CXCR4 receptor is critical to in the homing of BMSCs(Sordi, 2009; Jin et al., 2018; Hocking, 2015). For instance, one study reported that 43% ± 13% of human BMSCs (hBMSCs) express CXCR4 and are negative for CXCR3, which is consistent with our finding (Honczarenko et al., 2006). The lower expression of CXCR4 in hBMSCs relative to that in rBMSCs will be investigated in a follow-up study. In addition, the cells labeled with 25 µg/ml SPIOs for 48 h showed the highest expression of CXCR7 among all groups. However, there was no significant difference between the migration rates of the labeled and unlabeled rBMSCs, which suggests that the chemokine receptors affect the migration of rBMSCs by varying degrees. Given the higher expression of CXCR4 in the SPIOs-labeled cells compared to the controls, we surmise that it may be an important factor in the chemotactic migration of rBMSCs.

Interestingly, the labelling rate was higher in cells incubated with the SPIOs for 48 h compared to 24 h or 72 h. This is likely due to the fact that the number of cells accelerated rapidly during prolonged culture while the amount of SPIOs remained the same. The labeled cells may allocate the SPIOs into their descendant cells for several generations but after a certain threshold, the amount of SPIOs dilutes away. Thus, the percentage of labeled cells initially increased after 48 h and then decreased at 72 h regardless of the concentration of SPIOs. In addition, the concentration of SPIOs did not significantly affect the extent of labeling. However, it is possible that the actual concentration of iron particles in the rBMSCs treated with 75 µg/ml SPIOs may be higher than that in cells treated with 25 µg/ml SPIOs, although the percentage of labeled cells were similar in both groups. This hypothesis will be tested in future experiments. Furthermore, the viability of the cells incubated with SPIOs decreased in a time-dependent manner.

Foldager et al. (2011) had reported that very small iron oxide particles (VSOP) coated with citric acid downregulated the cartilage-specific sox9 and collagen type 2 genes in the chondrocytes. In contrast, we used larger iron particles without any coating, analyzed chemotaxis-related genes, and utilized a different cellular model. Xiaolei Huang et al. transduced BMSCs with lentiviruses expressing the ferritin heavy chain 1 (Fth 1) for in vivo MRI tracking, and found that while the MRI signal intensity of SPIO-BMSCs gradually reduced in a time-dependent manner, that of Fth 1-BMSCs were sustained for 10–60 days. Thus, SPIOs are steadily decomposed and excreted compared to Fth 1, which prevents iron overloading and the ensuing toxic effects (Huang et al., 2019).

There are several limitations in our study that ought to be considered. First, we did not analyze the secreted cytokine profile of the labeled BMSCs, which is of clinical relevance the cytokines secreted by the BMSCs influence hematopoiesis. Therefore, our next objective will be to analyze the mRNA and protein levels of multiple cytokines in the labeled rBMSCs. Furthermore, other genes involved in BMSC migration will also have to be analyzed. Finally, the clinical applicability of the SPIOs-labeled BMSCs will have to be validated in an in vivo aGVHD model. To summarize, our findings will help facilitate greater clinical use of Fe nanoparticles.

Conclusions

We have demonstrated for the first time that SPIOs can effectively label rBMSCs without affecting their viability and chemotactic functions in-vitro. While CXCR4 mRNA levels increased in the SPIO-labeled BMSCs, no changes were observed in the protein level. Furthermore, the SPIOs did not affect the chemotaxis of rBMSCs. The differences in the expression of CXCR4 mRNA and protein warrant further investigation. The in vivo homing of SPIO-labeled BMSCs can be safely monitored by MRI in order to assess their therapeutic effects against GVHD.

Supplemental Information

Supplemental Information 1 Raw data for Fig. 4

Click here for additional data file.

Supplemental Information 2 Raw data for Fig. 5

Click here for additional data file.

Supplemental Information 3 Raw data for Fig. 6

Click here for additional data file.

Supplemental Information 4 Raw data for Fig. 7

Click here for additional data file.

Supplemental Information 5 Author checklist

Click here for additional data file.

During the whole research, colleagues from the laboratory of Pediatrics and Gastroenterology of Guangzhou First People’s Hospital gave great support. Here we would like to thank their efforts.

Additional Information and Declarations

Competing Interests

Author Contributions

Data Availability

The authors declare there are no competing interests.

Yuanchun Liu conceived and designed the experiments, performed the experiments, analyzed the data, prepared figures and/or tables, authored or reviewed drafts of the article, and approved the final draft.

Wanyi Huang performed the experiments, prepared figures and/or tables, and approved the final draft.

Huiyang Wang performed the experiments, prepared figures and/or tables, and approved the final draft.

Wei Lu performed the experiments, prepared figures and/or tables, and approved the final draft.

Jiayu Guo analyzed the data, prepared figures and/or tables, and approved the final draft.

Li Yu analyzed the data, prepared figures and/or tables, and approved the final draft.

Lina Wang conceived and designed the experiments, analyzed the data, authored or reviewed drafts of the article, and approved the final draft.

The following information was supplied regarding data availability:

The raw measurements are available in the Supplemental Files.

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
