# Peer review of "Influence of SPIO labelling on the function of BMSCs in chemokine receptors expression and chemotaxis"

_PeerJ, doi:10.7717/peerj.15388_

## Round 0.1 · original submission · Major Revisions

Dear Dr. Liu,

Two independent reviewers have reviewed the manuscript and there are certain sections that require attention in the manuscript. Please address the comments raised by the reviewers and we look forward to the revised version of the manuscript.

Additionally there are multiple studies that have already reported the characterization and effect of superparamagnetic iron oxide nanoparticles and some of them have reported detailed characterization of labelling and characterized the function of mesenchymal stem cells in vitro and in vivo. The following manuscript are a few examples: PMID: 30666115,PMID: 28620162, PMID: 32152984,PMID: 25889904, PMID: 34511908. Can you please add a section in the revised manuscript discussing the significance of this study?

Please feel free to contact us if you have any further questions.

Best wishes.
Himangshu Sonowal

·

Basic reporting

1 - introduction (line 40): I would suggest adding some statistics (how widespread), description, and relevance of the complication mentioned, allo-HSCT.
2 - "graft versus host disease" should be corrected to "graft-versus-host disease" throughout the manuscript.
3 - lines 53-55: For context, it should be briefly discussed what the previous studies found about the toxic effects of labeling and imaging. Have there been any live animal studies testing effects on mortality? These questions are extremely important to ask before introducing or carrying out the work mentioned here. Hence, this context is needed in the manuscript.
4 - Lauridsen et al., Experimental and Therapeutic Medicine, 2018 is a crucial reference for this work, which is missing from the introduction and reference list.
5 - Several typos in Table 1 legend should be corrected.
6 - Figure 2: the legend seems incomplete: what is the difference between A and C?
7 - Figure 4: "rate" is misleading here. Usually rate refers to the time-differential, which has not been compared for the different experimental groups. Instead, the term "extent" seems more relevant.
8 - line 189: The result subtitle should be corrected to "SPIOs had no detrimental effect on in-vitro viability and proliferation of rBMSCs"
9 - Figure 5: the legend is missing information about how the expression levels were determined.
10- line 236: The result subtitle should include "in vitro" just like line the previous suggestion for line 189.
11- Figure 7: what is "Migrated number/LP" in the y-legend of B?
12- lines 258-261: the dose-effect of migration ability was not tested by the authors. Hence, this discussion point is misleading.

Experimental design

13- line 168: why are the percentages of CD** cells relevant? What information does that provide? i.e., why were these experiments conducted?
14- line 190: Why was tryptan blue dye exclusion tests performed? What information do these tests provide?
15- line 193: What is CCK-8 assay and why was it performed?
16- lines 206-207: Why were the mRNA levels of the mentioned genes measured? All the genes/proteins mentioned except IL11 seem to be genes encoding receptors but it needs to be explicitly mentioned in the manuscript. Are these all the known receptors? If not, why were these particular receptors chosen? This information is needed to provide context for the general audience.
17- line 214: Why was flow cytometry performed? What does the percentage of each receptor-type+ cells mean? Were the receptors overexpressed using a vector or does the + indicate that the receptors were inherently overexpressed as a result of the experimental test?
18- Negative controls are missing in the transwell assay (i.e., labeled and unlabeled cells with no attractant or repellent compounds)
19- line 247-248: what are TAK 799, MARAVIORIC and LY294002 and why are they relevant?

Validity of the findings

20- lines 33-35, lines 263-265, lines 298-299: the conclusion is not supported by the previous sentence or the experiments done in the manuscript. In-vitro functional analysis does not provide information about in vivo function or toxicity of any assay.
21- lines 33-35: the conclusion is not supported by the previous sentence or the experiments done in the manuscript. In-vitro functional analysis does not provide proof of in vivo function of any assay.
22- An earlier work in this field raised concerns that loading cells with huge amounts of iron can affect gene expression. This work, Foldager et al.,J Magn Reson Imaging, 2011, needs to be cited in the Introduction and its implication on the authors' present work needs to be discussed (either in Introduction or Discussion).
23- Can the authors comment how their SPIO labeling approach compares with Ferritin-based iron labeling? Huang and coworkers (Contrast media & molecular imaging, 2018) seem to suggest that MRI of SPIO labeling is not as robust as their Ferritin-based labeling approach. This reference is also missing from the literature cited by the authors.
24- lines 169-170: What are the previous reports and why are they supporting the statement made by the authors?
25- line 270: Is a higher level of chemokines healthy? A quick search suggests not. See examples: Hoffmann-Vold et al., Chest, 2016; Fig C in Roy et al., Surgery, 2014.

Additional comments

26- While the experimental methods are reported well by the authors, the context is severely lacking in two ways:
- The literature is not appropriately discussed, especially, more recent relevant works are missing from the literature cited. I would recommend the authors to carry out a literature review on recent works published on SPIO labeling as well as alternative techniques.
- The rationale for the experiments is not clear. This makes the results difficult to interpret.

27- Finally, the authors conclude direct application of the work to in-vivo studies, which is inaccurate and overreaching. This should be corrected in any updated versions of the manuscript.

Reviewer 2 ·

Basic reporting

The manuscript entitled "Influence of SPIO labelling on the function of BMSCs in chemokine receptors expression and chemotaxis" presented a clinical study to determine the impact of different concentrations and labelling durations of superparamagnetic iron oxide particles (SPIOs) on the biological characteristics. The authors have evaluated the effects of the SPIOs, specifically on the gene expression profile and chemotaxis the function of the BMSCs.
Pros: This is an exciting study, and the data are very informative about the consequence of SPIO labelling on the function of BMSCs, specifically on the chemokine expression. A comparable amount of different concentrations of SPIOs at various time points provides resilience to the manuscript. Overall, this is a concise, well-written, and good manuscript. The introduction is relevant and statics based. Adequate details regarding the previous studies' results and their association with the current manuscript are very well presented by the authors, and that makes the manuscript easy for readers to follow the present study rationale and procedures.

Experimental design

The material and methods are relevant; however, a few explanations for employing the approaches should be incorporated. Overall, the results are straightforward and persuade the view. Also, the authors have made an organized contribution to the research literature in this area of investigation; however, some recent studies should be cited and included in the current manuscript.

Validity of the findings

All the sections, including results, discussion, and conclusion, are well described, but including a few sentences in all the sections and adding details and references could improve it.

Additional comments

Though the manuscript is informative, a few sentences must be specific and provide detailed information. Also, a few references need to be included. In conclusion, the manuscript is suitable for publication after the authors have addressed the following comments and questions.

Annotated reviews are not available for download in order to protect the identity of reviewers who chose to remain anonymous.

---

## Round 0.2 · Minor Revisions

Dear Dr. Liu,

I am pleased to inform you that the revised version of the manuscript has been recommended for acceptance after a second round of review. However, there are some minor sections that require attention before final acceptance.

Please see my comments below:
1. Please make changes as suggested by the reviewers.
2. The references cited in the text need a careful review- For example: “Foldager et al” there is no reference number. There are multiple such sections throughout the manuscript. In addition, multiple sentences are quoted without corresponding references.
3. Editing throughout the manuscript is required: for example- density of 1*106 cells/tube
4. Materials and methods-Flow cytometry plots showing percentages of (A) CD29+, (B) CD44+, (C) CD34- and (D) CD45- cells-Please explain clearly how the gates for determining the % positive cells were chosen?
5. Scale bars in images are required.
6. Gene names, symbols (eg. microgram/ml is not "u/ml"), using * where not necessary etc. please correct throughout the and follow standard scientific nomenclature.

Best,

Himangshu Sonowal
Editor PeerJ

·

Basic reporting

- The authors have satisfactorily addressed the reviewer comments.

- In the newly added text, the authors need to double-check the citations: e.g., in line 323, work by Foldager et al. is mentioned but this sentence does not have a numeric reference to the article. Similarly for Xiaolei Huang et al. in line 326. These are just examples, the authors should check throughout the manuscript.

- line 252: "rate" should be changed to "extent".

Experimental design

The manuscript now explains the motivations for experimental designs.

Validity of the findings

The authors have satisfactorily resolved the queries by discussing past literature.

Additional comments

Minor comment: The spelling of labeling/labelling is inconsistent throughout the manuscript. This should be fixed.

Reviewer 2 ·

Basic reporting

I have reviewed the manuscript, and the authors have incorporated all the concerns and suitable references. The manuscript can be accepted in its current form.

Experimental design

I have reviewed the manuscript, and the authors have incorporated all the concerns and suitable references. The manuscript can be accepted in its current form.

Validity of the findings

I have reviewed the manuscript, and the authors have incorporated all the concerns and suitable references. The manuscript can be accepted in its current form.

Additional comments

I have reviewed the manuscript, and the authors have incorporated all the concerns and suitable references. The manuscript can be accepted in its current form.

---

## Round 0.3 · accepted · Accept

Dear Dr. Liu,

Thank you for your submission to PeerJ. The manuscript is now formally accepted for publication. Congratulations!

Best,
Himangshu Sonowal
Academic Editor
PeerJ Life & Environment